EMBO
Molecular Medicine

# Defective PITRM1 mitochondrial peptidase is associated with Aβ amyloidotic neurodegeneration

Dario Brunetti[1,†], Janniche Torsvik[2,†], Cristina Dallabona[3], Pedro Teixeira[4], Pawel Sztromwasser[5,6], Erika Fernandez-Vizarra[1], Raffaele Cerutti[1], Aurelio Reyes[1], Carmela Preziuso[7], Giulia D'Amati[7], Enrico Baruffini[3], Paola Goffrini[3], Carlo Viscomi[1], Ileana Ferrero[3], Helge Boman[8], Wenche Telstad[9], Stefan Johansson[5,8], Elzbieta Glaser[4], Per M Knappskog[5,8], Massimo Zeviani[1,*] & Laurence A Bindoff[2,10,**]

## Abstract

Mitochondrial dysfunction and altered proteostasis are central features of neurodegenerative diseases. The pitrilysin metallopeptidase 1 (PITRM1) is a mitochondrial matrix enzyme, which digests oligopeptides, including the mitochondrial targeting sequences that are cleaved from proteins imported across the inner mitochondrial membrane and the mitochondrial fraction of amyloid beta (Aβ). We identified two siblings carrying a homozygous PITRM1 missense mutation (c.548G>A, p.Arg183Gln) associated with an autosomal recessive, slowly progressive syndrome characterised by mental retardation, spinocerebellar ataxia, cognitive decline and psychosis. The pathogenicity of the mutation was tested in vitro, in mutant fibroblasts and skeletal muscle, and in a yeast model. A Pitrm1[+/−] heterozygous mouse showed progressive ataxia associated with brain degenerative lesions, including accumulation of Aβ-positive amyloid deposits. Our results show that PITRM1 is responsible for significant Aβ degradation and that impairment of its activity results in Aβ accumulation, thus providing a mechanistic demonstration of the mitochondrial involvement in amyloidotic neurodegeneration.

**Keywords** amyloid beta; mitochondrial targeting sequence; mitochondrial disease; neurodegeneration; pitrilysin 1
**Subject Categories** Genetics, Gene Therapy & Genetic Disease; Metabolism; Neuroscience

See also: **V Boczonadi & R Horvath** (March 2016)

## Introduction

Mitochondrial dysfunction, whether primary or secondary, is increasingly recognised as a hallmark of neurodegeneration (Johri & Beal, 2012; Moran et al, 2012). Not only is the brain a major target in primary, genetically determined mitochondrial disease, but mitochondrial dysfunction is also a prominent feature in many of the most prevalent neurodegenerative diseases including Parkinson's disease (PD) and Alzheimer's dementia (AD) (Manczak et al, 2006; Morais & De Strooper, 2010; Friedland-Leuner et al, 2014). For instance, AD is characterised by the accumulation of the amyloid beta (Aβ) peptide as plaques in the neuropil, and recent work has suggested that Aβ is present in the inner compartment of mitochondria (Falkevall et al, 2006; Manczak et al, 2006; Hansson Petersen et al, 2008). The mitochondrial fraction of Aβ is quantitatively digested by the pitrilysin metallopeptidase 1 (PITRM1) (Hansson Petersen et al, 2008; Pagani & Eckert, 2011; Pinho et al, 2014). PITRM1 (also known as presequence peptidase, PreP) is a 117 kDa mitochondrial matrix enzyme. In addition to its role in the disposal of mitochondrial Aβ, PITRM1 is deemed responsible for digesting the mitochondrial targeting sequence (MTS) of proteins imported across the inner mitochondrial membrane (Stahl et al, 2002; Alikhani et al, 2011a,b; Teixeira & Glaser, 2013), which are cleaved from the mature polypeptides by the mitochondrial matrix peptidase (MMP). Interestingly, the accumulation of Aβ peptides has been shown to inhibit the activity of Cym1, the PITRM1 orthologue in yeast, leading to impaired MTS processing

1 MRC Mitochondrial Biology Unit, Wellcome Trust, Cambridge, UK
2 Department of Neurology, Haukeland University Hospital, Bergen, Norway
3 Department of Life Sciences, University of Parma, Parma, Italy
4 Department of Biochemistry and Biophysics, Stockholm University, Stockholm, Sweden
5 Department of Clinical Science, University of Bergen, Bergen, Norway
6 Computational Biology Unit, Department of Informatics, University of Bergen, Bergen, Norway
7 Department of Radiological, Oncological and Pathological Sciences, Sapienza University of Rome, Rome, Italy
8 Center for Medical Genetics and Molecular Medicine, Haukeland University Hospital, Bergen, Norway
9 Department of Neurology, Førde Hospital, Førde, Norway
10 Department of Clinical Medicine (K1), University of Bergen, Bergen, Norway
*Corresponding author. Tel: +44 1223 252704; Fax: +44 1223 252715; E-mail: mdz21@mrc-mbu.cam.ac.uk
**Corresponding author. Tel: +47 55975096; Fax: +47 55975164; E-mail: laurence.bindoff@nevro.uib.no
†These authors contributed equally to this work

and accumulation of precursor proteins (Mossmann *et al*, 2014). We report here a family carrying a missense mutation in *PITRM1* associated with a slowly progressive neurodegenerative phenotype. Investigations *in vitro* as well as in yeast and mouse models allowed us to clarify the mechanism of the disease and shed new light on the relationship between mitochondria and neurodegeneration.

# Results

### PITRM1 is mutated in patients with a neurodegenerative phenotype

We studied a single index family (Fig 1A) coming from a small Norwegian coastal community comprising < 200 individuals. Of five siblings, two are definitely affected (II-2, II-4); one unaffected sibling has a peripheral neuropathy (II-1); another has psychiatric symptoms, but refuses investigation (II-5); and one died of cancer (II-3) before we ascertained the family. Our study was approved by the Regional Committee for Medical and Health Research Ethics, Western Norway. The index case (II-2), now 68 years old, was diagnosed as a child with mild mental retardation and later developed gradual spinocerebellar ataxia (SCA), obsessional behaviour with psychotic episodes and hallucinations. Brain MRI (Fig 1B) showed marked cerebellar atrophy and unilateral signal changes in the thalamus. Routine blood profile was unremarkable, but CSF examination showed low $A\beta_{1-42}$ (363 ng/l; n.v. > 550), similar to that seen in idiopathic AD (Motter *et al*, 1995; Andreasen *et al*, 1999). Total and phosphorylated Tau and 14-3-3 proteins were normal. A muscle biopsy showed some scattered COX-negative fibres (Fig 1C). Her brother (II-4) was also described as mildly mentally retarded from an early age, had obsessional behaviour and episodes of psychosis, and early onset ataxia. His CT scan showed cerebellar and some cerebral atrophy. Respiratory chain (RC) complex assays in muscle homogenate from individual II-2 showed low specific activities of all complexes, and a concomitant decrease in citrate synthase (CS), an index of mitochondrial mass (Fig 1D). When specific activities were normalised to CS, no significant changes were

observed relative to controls, except for a trend towards complex I decrease. No abnormality was seen in an enriched mitochondrial fraction from mutant immortalised fibroblasts. Combined homozygosity SNP-based mapping and whole exome sequencing (WES) were carried out in II-2 based on an autosomal recessive mode of inheritance. WES resulted in 20,436 genetic variants, of which 240 were coding and not found in our in-house frequency database or in the 1000 Genomes database at > 0.5% allele frequency. A total of 13 genes contained rare variants consistent with autosomal recessive inheritance, and of these, only one gene, *PITRM1*, was located within a homozygous region on chromosome 10p shared by both affected siblings. The mutation NM_014889.2: c.548G>A, verified by Sanger sequencing (Appendix Fig S1), was absent in the clinically unaffected brother (II-1), in > 300 normal control individuals from Western Norway and in the ExAc database (http://exac.broadinstitute.org). It predicts the synthesis of a p.R183Q variant. The R183 residue is conserved in both humans and baker's yeast.

### *PITRM1*[R183Q] is unstable and impairs mitochondrial function

To evaluate the pathogenic effect of the p.R183Q mutation, we first investigated skin fibroblasts and a skeletal muscle biopsy taken from subject II-2.

*PITRM1* RNA expression measured by qPCR was similar in *PITRM1*[R183Q] vs. *PITRM1*[wt] cells (not shown). However, Western blot analysis of proteins separated by SDS–PAGE showed marked reduction of PITRM1 amount in II-2 fibroblasts and skeletal muscle (Fig 1E), suggesting protein instability.

To test the effect of the PITRM1[R183Q] mutation on catalytic activity, we expressed recombinant PITRM1[R183Q] and PITRM1[wt] in *Escherichia coli*. Both affinity-purified protein variants showed equal ability to cleave either of two fluorogenic peptides (Fig 1F), or Aβ (Fig 1G). Taken together, these results demonstrate that PITRM1[R183Q] is catalytically active *in vitro*, but highly unstable *in vivo*.

To characterise the cellular pathophysiology associated with PITRM1[R183Q], we studied immortalised fibroblasts from subject II-2. These cells showed a significant growth defect on respiration-obligatory galactose medium, but not on glycolytic-permissive glucose

---

**Figure 1. Clinical and molecular studies on *PITRM1*[R183Q] mutant patients.**

A   Family tree. Affected subjects are represented by filled shapes. The index case is II-2 (arrow).

B   Brain MRI images of II-1 (FLAIR). A: (a) sagittal interemispheric sequence showing profound cerebellar atrophy (arrow); (b) transverse sequence showing involvement of the dentate nuclei of cerebellum; (c) coronal sequence revealing cortical atrophy, a wide third ventricle, and high signal changes in the thalami, particularly on the left (arrow); (d) the thalamic lesion is indicated by an arrow.

C   Histochemical staining of skeletal muscle from II-2. Combined COX/SDH staining shows scattered COX-negative (blue) fibres (arrows). Scale bar corresponds to 100 μm.

D   Respiratory chain complex activities in skeletal muscle. Specific activities (nmol/min/mg) of complex I, II and IV are reduced in II-2 (red) compared to the controls mean (blue); CS activity is also low (< 50% of controls), suggesting a reduction in mitochondrial mass (see main text for further details). Each activity was measured in triplicate.

E   Western blot analysis of PITRM1 in primary fibroblasts (left) and skeletal muscle (right) of controls (CTR) and subject II-2. Densitometric quantification using the Genetools software is shown below the blots. In blue is PITRM1[WT] and in red PITRM1[R183Q].

F   Proteolytic activity on two fluorescent oligopeptides (AnaSpec), reported at the bottom of each histogram, by 6-His-tagged PITRM1[WT] and PITRM1[R183Q] proteins expressed in *E. coli* and affinity-purified by Ni-agarose chromatography. Values are expressed as arbitrary units per sec per μg of protein (a.u./s/μg). Experiments were performed in duplicate.

G   Degradation rate of $A\beta_{1-42}$ by purified 6-His-tagged PITRM1[WT] and PITRM1[R183Q] proteins. Quantification of these experiments is displayed below the blots. Experiments were performed in duplicate.

Data information: Data are presented as mean ± SD.
Source data are available online for this figure.

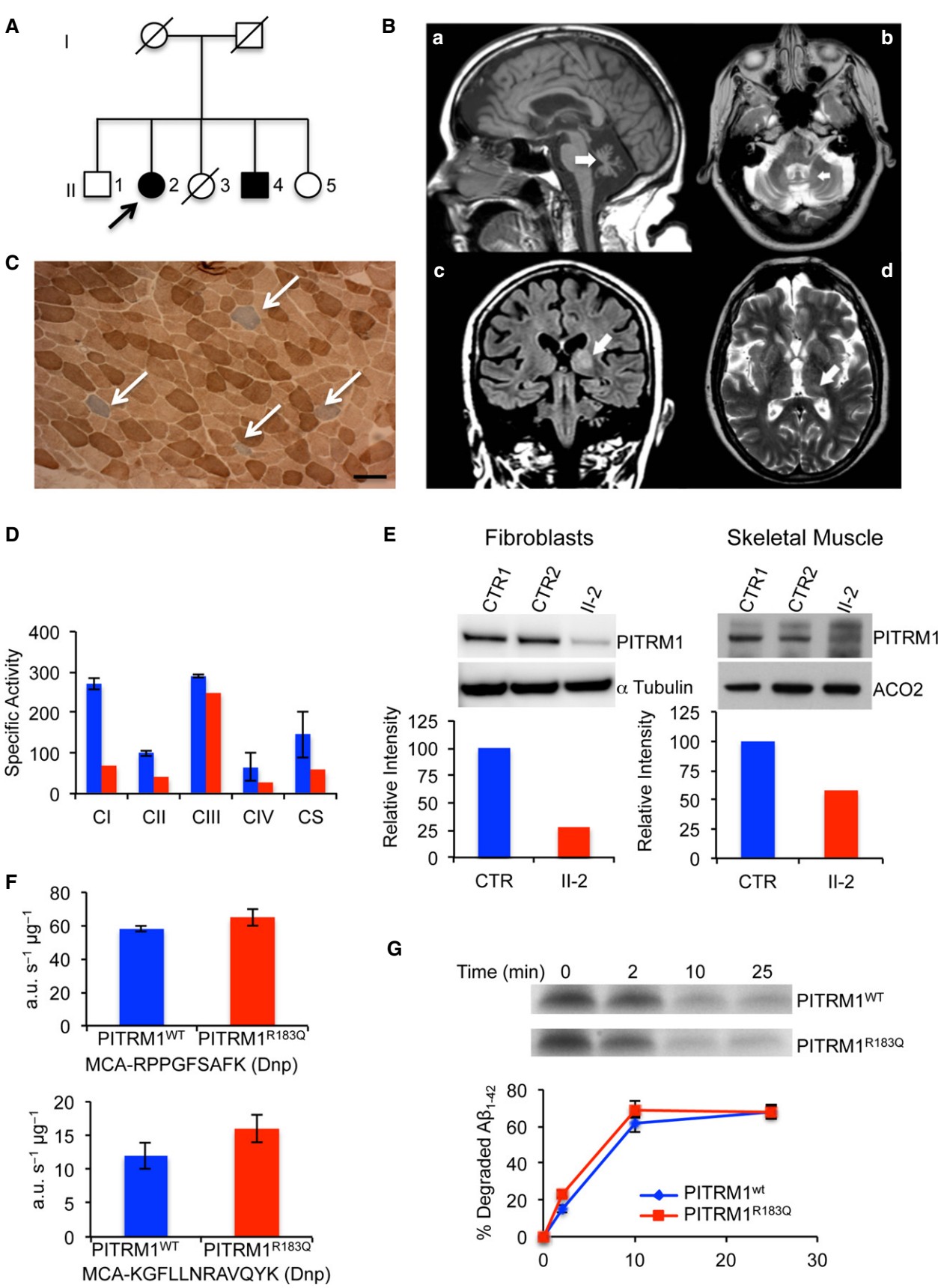

Figure 1.

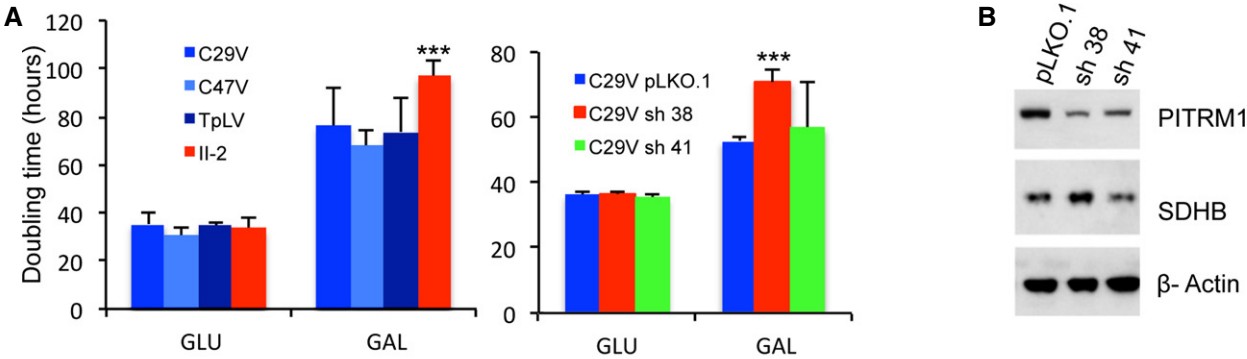

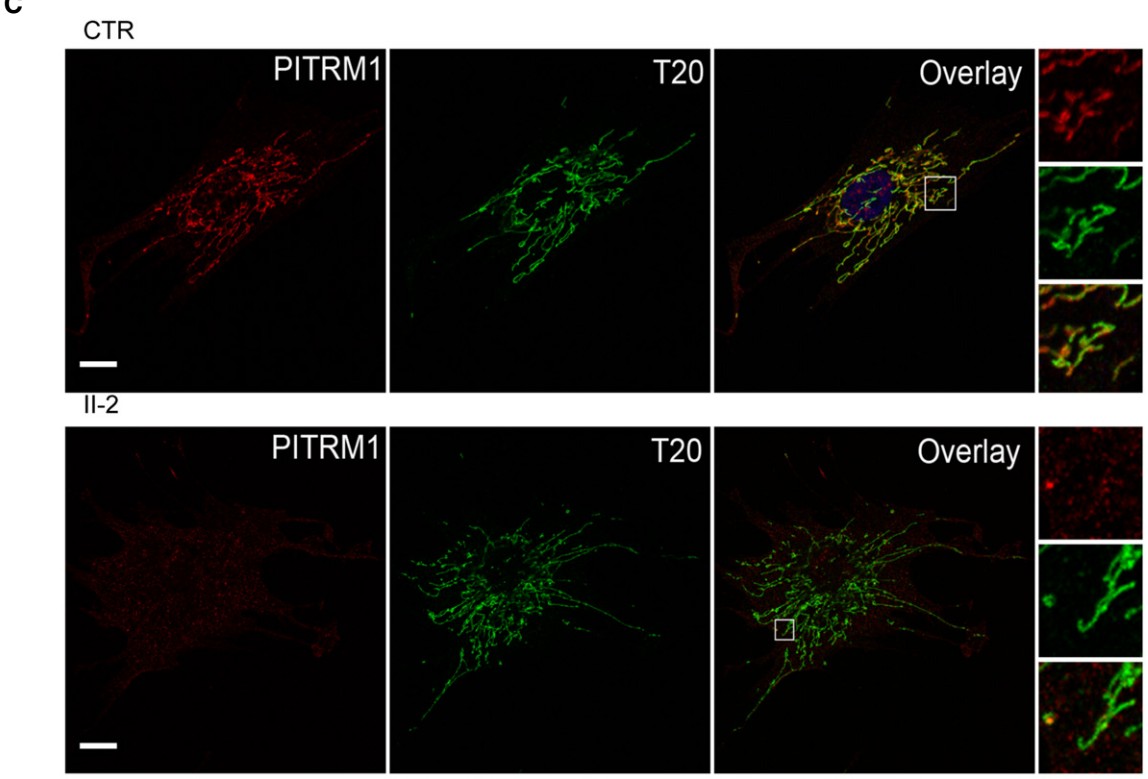

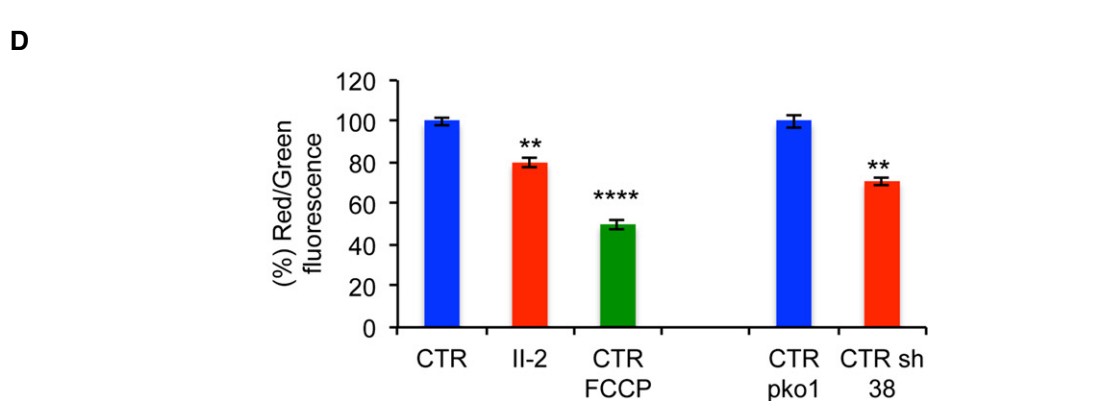

**Figure 2.**

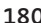

**Figure 2.  Characterisation of immortalised cells.**

A  Cell growth in glucose (GLU) and galactose (GAL). C29V, C47V and TpLV: immortalised fibroblast cell lines from control individuals; PITRM1V: immortalised fibroblasts from subject II-2, carrying the $PITRM1^{R183Q}$ mutation. pLKO.1: empty vector; sh38: shRNA38; sh41: shRNA41. Each cell line was measured six times. Statistical analysis was performed using two-way ANOVA *post hoc* Bonferroni test, ***$P$ < 0.001.

B  Western blot analysis of PITRM1 in C29V cells transduced with the empty vector pIKO.1 and the two shRNAs, sh38 and sh41. SDHB and β-actin are used as loading controls.

C  Co-localisation of PITRM1 (red) with TOM20 (T20, green) in human fibroblast cells from a control (CTR) and subject II-2. Note that the intensity of PITRM1 immunofluorescence is much lower in II-2 cells than in CTR cells (see main text for further details). Nuclei are stained in blue by DAPI. Scale bar corresponds to 10 μm.

D  Mitochondrial membrane potential (Δ$P$). In healthy cells with high mitochondrial Δ$P$, JC-1 forms complexes known as J-aggregates that show an intense red fluorescence. On the contrary in unhealthy cells with low Δ$P$, JC-1 remains in the monomeric form, showing only green fluorescence. Values are referred to as % of those of the control means, taken as 100%. The results are the mean of three independent experiments. Statistical analysis was performed using two-way ANOVA, **$P$ < 0.01; ****$P$ < 0.0001.

Data information: Data are presented as mean ± SD. Exact $P$-values are reported in Table EV1.
Source data are available online for this figure.

medium, compared to PITRM1$^{wt}$ cells (Fig 2A). Interestingly, similar results were obtained in control immortalised fibroblasts stably expressing a *PITRM1*-specific shRNA, which decreased its protein levels to approximately 40% of the amount found in cells transfected with the empty vector (Fig 2A and B). The mitochondrial localisation of PITRM1 was confirmed using immunofluorescence (Fig 2C). Mitochondrial DNA (mtDNA) levels were similar in PITRM1$^{R183Q}$ vs. PITRM1$^{wt}$ cells, and there was no evidence of mtDNA depletion or significant amount of multiple deletions (not shown).

The mitochondrial membrane potential (Δ$P$) was significantly lower in PITRM1$^{R183Q}$ vs. PITRM1$^{wt}$, but higher than that measured in PITRM1$^{wt}$ cells treated with the Δ$P$ dissipator carbonyl cyanide-p-trifluoromethoxyphenylhydrazone (FCCP). Likewise, Δ$P$ was significantly lower than normal in shRNA-silenced PITRM1$^{wt}$ cells (Fig 2D).

In order to further ascertain the functional significance of $PITRM1^{R183Q}$, we modelled the mutation in the yeast *Saccharomyces cerevisiae*, whose *PITRM1* orthologue is *CYM1* (Alikhani *et al*, 2011a). The *CYM1* null mutant, *cym1Δ*, was transformed with the $cym1^{R163Q}$ allele, carrying a mutation equivalent to that found in our family.

When cultures were grown at standard temperature (28°C), hardly any difference was observed (Appendix Fig S2). At 37°C, however, the *cym1Δ* strain displayed impaired oxidative growth (Fig 3A), the O$_2$ consumption rate (OCR) was as low as 25% of the wild-type (Fig 3B), and cytochromes content was profoundly reduced (Fig 3C). All subsequent experiments were, therefore, carried out at 37°C. In these conditions, the transformation of the *cym1Δ* strain with *CYM1*$^{wt}$ restored OCR and cytochrome content, whereas transformation with the $cym1^{R163Q}$ variant gave intermediate but clearly defective results (Fig 3A–C). The enzymatic activities of complexes II, III and IV of the yeast strains paralleled the results of the OCR (Fig 3D). Western blot analysis of the $cym1^{R163Q}$ yeast strain showed reduced amount of cym1 protein compared to the *CYM1*$^{wt}$ strain, whereas no protein was present in the *cym1Δ* strain, as expected (Fig 3E). By overexposing the blot (Fig 3F), a low molecular weight band (arrow) was consistently observed below the $cym1^{R163Q}$ band, likely corresponding to a cym1$^{R163Q}$ degradation product. These results indicate that the cym1$^{R163Q}$ mutant protein is unstable and prone to accelerated degradation, similar to the human PITRM1$^{R183Q}$ mutant protein.

Finally, to test whether the $cym1^{R163Q}$ mutation affected the stability of mitochondrial Aβ$_{1-42}$, we expressed a modified Aβ$_{1-42}$ fused with the mitochondrial signal peptide of Sod2. This fusion protein is targeted to mitochondria and could be identified by an anti Aβ$_{1-42}$ antibody. Whilst this Aβ$_{1-42}$ was almost completely eliminated in the *CYM1*$^{wt}$, no degradation occurred in the *cym1Δ* strain, and incomplete digestion of Aβ$_{1-42}$ was observed in the $cym1^{R163Q}$ strain (Fig 3G). Taken together, our results in yeast demonstrate the pathogenic role of the $cym1^{R163Q}$ mutation equivalent to human $PITRM1^{R183Q}$.

**Figure 3.  Modelling the $cym1^{R163Q}$ mutation in *Saccharomyces cerevisiae*.**

A  Oxidative growth. W303-1B *cym1Δ* strains harbouring the wild-type *CYM1* allele (*CYM1*$^{wt}$), the $cym1^{R163Q}$ mutant allele or the empty vector were serially diluted from $10^7$ to $10^4$ cells/ml. Five microlitres of each dilution was spotted on SC agar plates without uracil, supplemented with 2% glucose, 2% glycerol or 2% ethanol. Plates were incubated at 37°C for 3–7 days.

B  Oxygen consumption rate (OCR). Cells grown at 37°C SC medium without uracil were supplemented with 0.5% glucose. Values were normalised to the OCR of the *CYM1*$^{wt}$ strain (49 nmol O$_2$/min/mg) and represented as the mean of at least three values ± SD.

C  Reduced versus oxidised cytochrome spectra. Peaks at 550, 560 and 602 nm correspond to cytochromes *c*, *b* and *aa3*, respectively. The height of each peak relative to the baseline is an index of cytochrome content.

D  Respiratory chain complex activities. Biochemical activities of succinate quinone DCPIP reductase, SQDR (CII), NADH-cytochrome *c* oxidoreductase activity NCCR (CIII) and cytochrome *c* oxidase (CIV) were measured on a mitochondrial enriched fraction from cells grown at 37°C as in (B). Values were normalised to that of *CYM1*$^{wt}$ strain and represented as the mean of three independent experiments ± SD.

E  Western blot on total protein extract using an anti-HA monoclonal antibody visualising the CYM1$^{wt}$ and cym1$^{R163Q}$ recombinant proteins both fused in frame with the HA epitope on the C-terminus. Total protein extracts were obtained by strains expressing HA-tagged *CYM1*$^{wt}$ and $cym1^{R163Q}$. PGK was used as a loading control, and signals were normalised to the wt. The quantification was performed on five independent blots.

F  Prolonged exposure of a Western blot containing the CYM1$^{wt}$ and cym1$^{R163Q}$ recombinant proteins reveals the presence of a band corresponding to a degradation product in the cym1$^{R163Q}$ lanes (arrow).

G  Western blot of Aβ$_{1-42}$$^{myc}$ monomer and dimer incubated with purified mitochondrial extracts from cells grown at 37°C in SC medium supplemented with 0.15% glucose and 2% galactose. VDAC was used as a loading control. Each experiment was performed in triplicate.

Data information: Statistical analysis was performed using unpaired, two-tailed Student's *t*-test. **$P$ < 0.01; ***$P$ < 0.001. Exact $P$-values are reported in Table EV1.
Source data are available online for this figure.

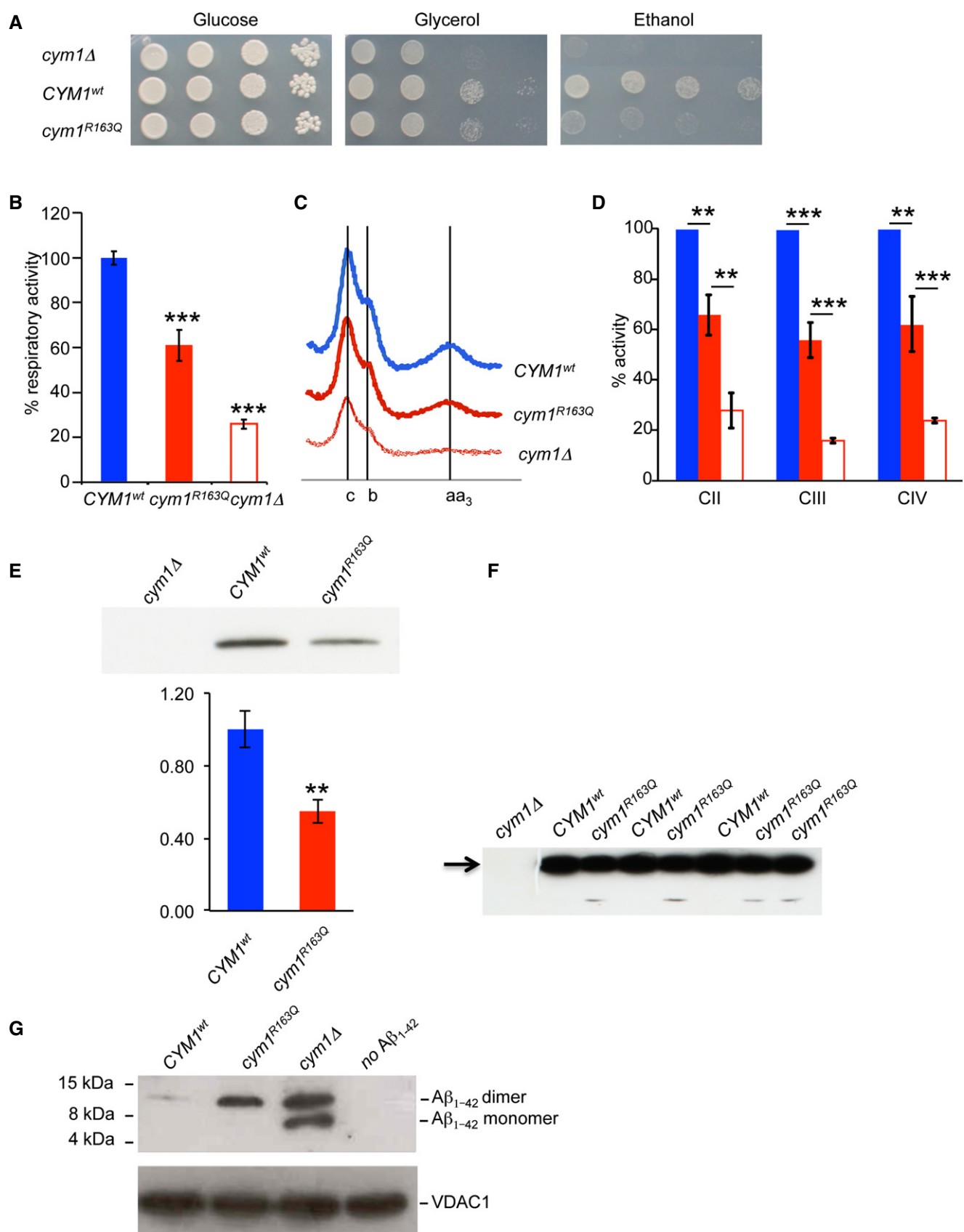

**Figure 3.**

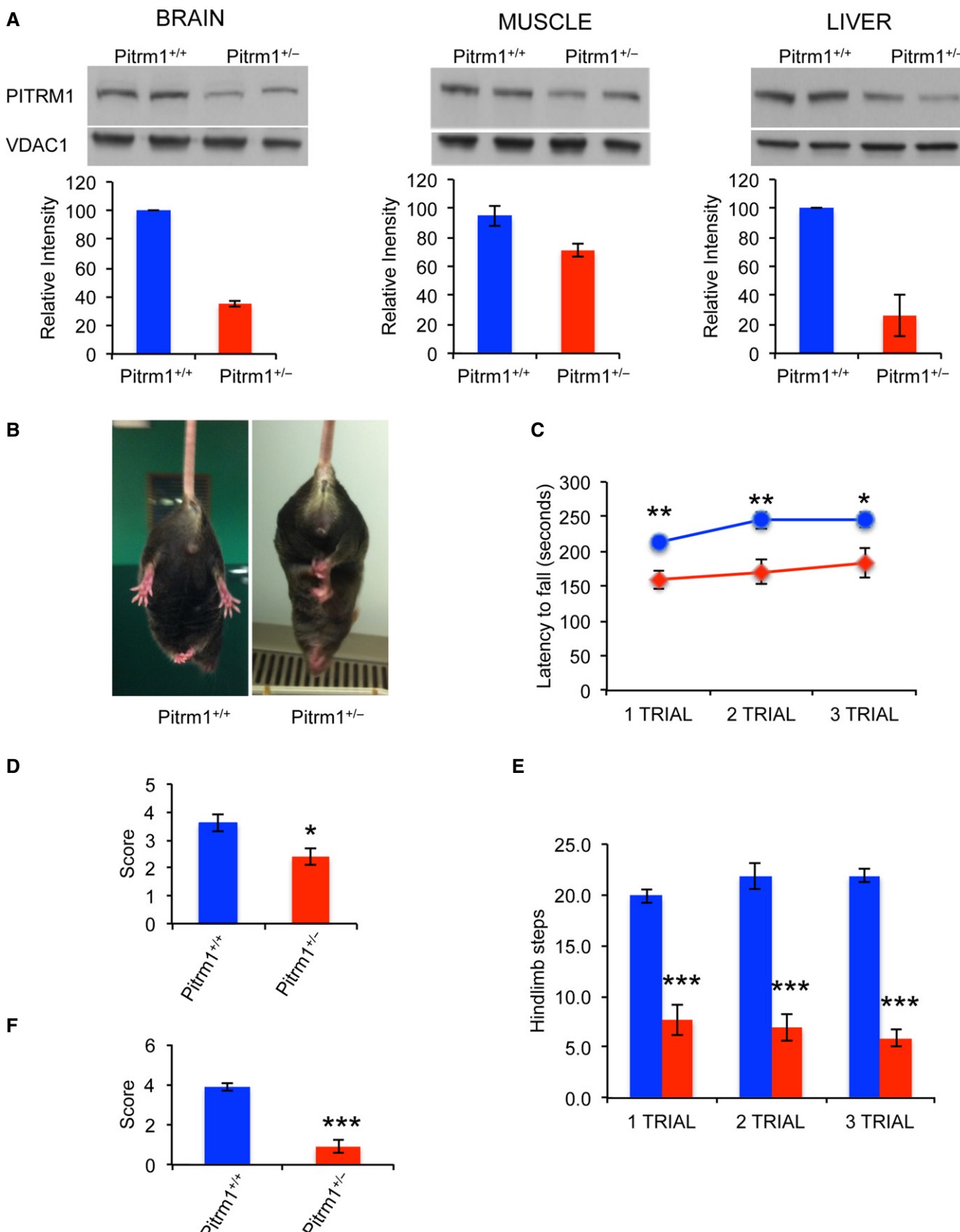

Figure 4.

**Figure 4. Behavioural studies of *Pitrm1*$^{+/-}$ mouse.**

A    Western blot analysis of Pitrm1 protein in brain, muscle and liver of two 4-mo male *Pitrm1*$^{+/-}$ mice and two *Pitrm1*$^{+/+}$ littermates. Densitometric analysis is reported in the histograms below the blots. *Pitrm1*$^{+/+}$ is in blue and *Pitrm1*$^{+/-}$ in red.

B    Representative hindlimb clasping phenomenon is shown in a 4-mo *Pitrm1*$^{+/-}$ male mouse, consisting in strong adduction of the hindlimbs when the animal is suspended by the tail; a littermate *Pitrm1*$^{+/+}$ control displays the normal reflex, consisting in wide abduction of the limbs. All examined *Pitrm1*$^{+/-}$ animals displayed this abnormal reflex from 2 months of age.

C    Rotarod test. Blue and red lines refer to *Pitrm1*$^{+/+}$ (n = 9) and *Pitrm1*$^{+/-}$ (n = 7) 4-mo animals, respectively.

D–F    Negative geotaxis (D), cylinder (E) and pole (F) tests. The experiments were carried out on the same group of animals as in (B).

Data information: Data are presented as mean ± SD. Statistical analysis was performed using unpaired, two-tailed Student's *t*-test. *$P < 0.05$; **$P < 0.01$; ***$P < 0.001$. Exact *P*-values are reported in Table EV1.

Source data are available online for this figure.

## *Pitrm1*$^{+/-}$ mice develop a neurodegenerative phenotype, with accumulation of Aβ$_{1-42}$ and signal peptides

Next, a *Pitrm1* knock-out C57BL/6n-A$^{tm1Brd}$ mouse line was obtained from the Wellcome Trust Sanger Institute, Cambridge, UK. Whilst the constitutive *Pitrm1*$^{-/-}$ genotype is associated with embryonic lethality, *Pitrm1*$^{+/-}$ heterozygotes survive to adulthood. In 4-month-old (mo) *Pitrm1*$^{+/-}$, Pitrm1 levels were well under 50% in brain and liver, and approximately 60% in skeletal muscle, compared to *Pitrm1*$^{+/+}$ littermates (Fig 4A), thus replicating the molecular lesion found in *PITRM1*$^{R183Q}$ patients (reduced amount of a catalytically normal enzyme). Mice were monitored weekly for onset of postural abnormalities, weight loss and general health. No significant weight differences were found between the two groups. The first evidence of abnormality was the development of hindlimb clasping in heterozygous *Pitrm1*$^{+/-}$ male mice from the age of 2 months. The neurological phenotype was evaluated further with a set of different coordination and sensorimotor tests in 4-mo males. The *Pitrm1*$^{+/-}$ heterozygous mice (n = 7) showed hindlimb clasping reflex (Brunetti *et al*, 2014) (Fig 4B) and performed poorly on tests of coordination, that is rotarod (Hickey *et al*, 2005), (Fig 4C) and negative geotaxis (Rogers *et al*, 1997) (Fig 4D); spontaneous rearing, that is cylinder test (Fleming *et al*, 2004) (Fig 4E); and basal ganglia-related movement control, that is pole test (Fig 4F). Metabolic assessment showed significantly reduced O$_2$ consumption and CO$_2$ production, and reduced heat production over 36 h of observation in animals housed in a comprehensive laboratory animal monitoring system (CLAMS, Columbus Instruments, Columbus, OH, USA) (Appendix Fig S3). Next, we carried out post-mortem analysis of *Pitrm1*$^{+/-}$ vs. *Pitrm1*$^{+/+}$ mice. We first performed Western blot analysis on 4-mo brain homogenates, using an antibody specific against the amyloid precursor protein (APP) and observed an approximately 2.5-fold increase of APP cross-reacting material in *Pitrm1*$^{+/-}$ vs. *Pitrm1*$^{+/+}$ specimens (Fig 5A). This result was concordant with immunohistochemical analysis of formalin-fixed and paraffin-embedded brains of same age, using the same anti-APP antibody (Appendix Fig S4A). In order to characterise the neuropathology of our *Pitrm1*$^{+/-}$ animals, we then carried out histological and immunohistochemical analysis on brain tissue. Three brains from 6-mo male *Pitrm1*$^{+/-}$ mice showed normal histochemical reactions to COX and SDH, two respiratory chain activities (not shown), and were then analysed for amyloid detection on formalin-fixed, paraffin-embedded specimens, using Congo red and Thioflavin T stainings. Congo red staining was viewed under polarised light, while Thioflavin T staining was evaluated by fluorescence microscopy. This analysis revealed scattered Thioflavin

and Congo red-positive areas, indicating the presence of amyloid deposits (Fig 5B). The Congo red-positive areas also showed apple-green birefringence under polarised light, another characteristic reaction of amyloid. Finally, the presence of immunofluorescence-positive areas was confirmed using an anti Aβ$_{1-42}$ antibody. Similar findings were also obtained in 4-mo and 12-mo male *Pitrm1*$^{+/-}$ animals (not shown). Additionally, 12-mo *Pitrm1*$^{+/-}$ brains underwent systematic histological and immunohistochemical analysis. *Pitrm1*$^{+/-}$ specimens showed increased gliosis (Appendix Fig S4B), and accumulation of ubiquitin-positive material in the neuropil and neurons (Fig 5B), with mild neuronal loss (not shown). The number of Aβ$_{1-42}$-immunoreactive areas was increased in 12-mo *Pitrm1*$^{+/-}$ brains, suggesting age-dependent accumulation (Fig EV1). No such lesions were ever found in any brain specimen from *Pitrm1*$^{+/+}$ littermates. In *Pitrm1*$^{+/-}$ female individuals of various ages, we found essentially the same clinical and neuropathological alterations as in males (not shown), indicating that the neurodegenerative process of the mouse model is not gender specific.

To further test the mechanistic consequences of impaired Pitrm1 activity on mitochondrial Aβ disposal, we then performed a time course for the import of Aβ$_{1-42}$ into isolated mitochondria from *Pitrm1*$^{+/+}$ and *Pitrm1*$^{+/-}$ mice. Isolated liver and brain mitochondria were incubated with Aβ$_{1-42}$ for 5 to 90 min. The levels of Aβ$_{1-42}$ interacting with mitochondria increased with incubation time in liver and brain, in both *Pitrm1*$^{+/+}$ and *Pitrm1*$^{+/-}$ mice (Fig 6A). However, in trypsin-digested samples, where the extra-mitochondrial proteins had been eliminated, Aβ$_{1-42}$ was decreasing over time in *Pitrm1*$^{+/+}$, whereas it accumulated in *Pitrm1*$^{+/-}$, clearly indicating impaired Aβ$_{1-42}$ degradation rate. Hence, we performed an Aβ$_{1-42}$ chase experiment by incubating the mitochondria with Aβ$_{1-42}$ for 15 min followed by treatment with trypsin and extensive washing. Imported Aβ$_{1-42}$ was almost fully degraded within 30 min of chase in both liver and brain from *Pitrm1*$^{+/+}$ mice, whereas a significant amount of Aβ$_{1-42}$ was still detected after 90 min of chase in both *Pitrm1*$^{+/-}$ tissues (Fig 6B). These results demonstrate limited capacity of *Pitrm1*$^{+/-}$ brain and liver mitochondria to eliminate Aβ$_{1-42}$, causing this peptide to accumulate.

Since PITRM1 is also responsible for digesting the MTS of proteins imported across the inner mitochondrial membrane (Stahl *et al*, 2002; Alikhani *et al*, 2011a,b; Teixeira & Glaser, 2013), we analysed the levels of a MTS in *Pitrm1*$^{+/+}$ and *Pitrm1*$^{+/-}$ mice. TFAM is a mitochondrial matrix protein, whose MTS serves as a substrate of PITRM1, whereas MPV17 is an inner membrane bound protein, which is not cleaved upon import into mitochondria. Radio-labelled TFAM was imported in a time-dependent fashion and at the same levels in *Pitrm1*$^{+/+}$ and *Pitrm1*$^{+/-}$ liver mitochondria

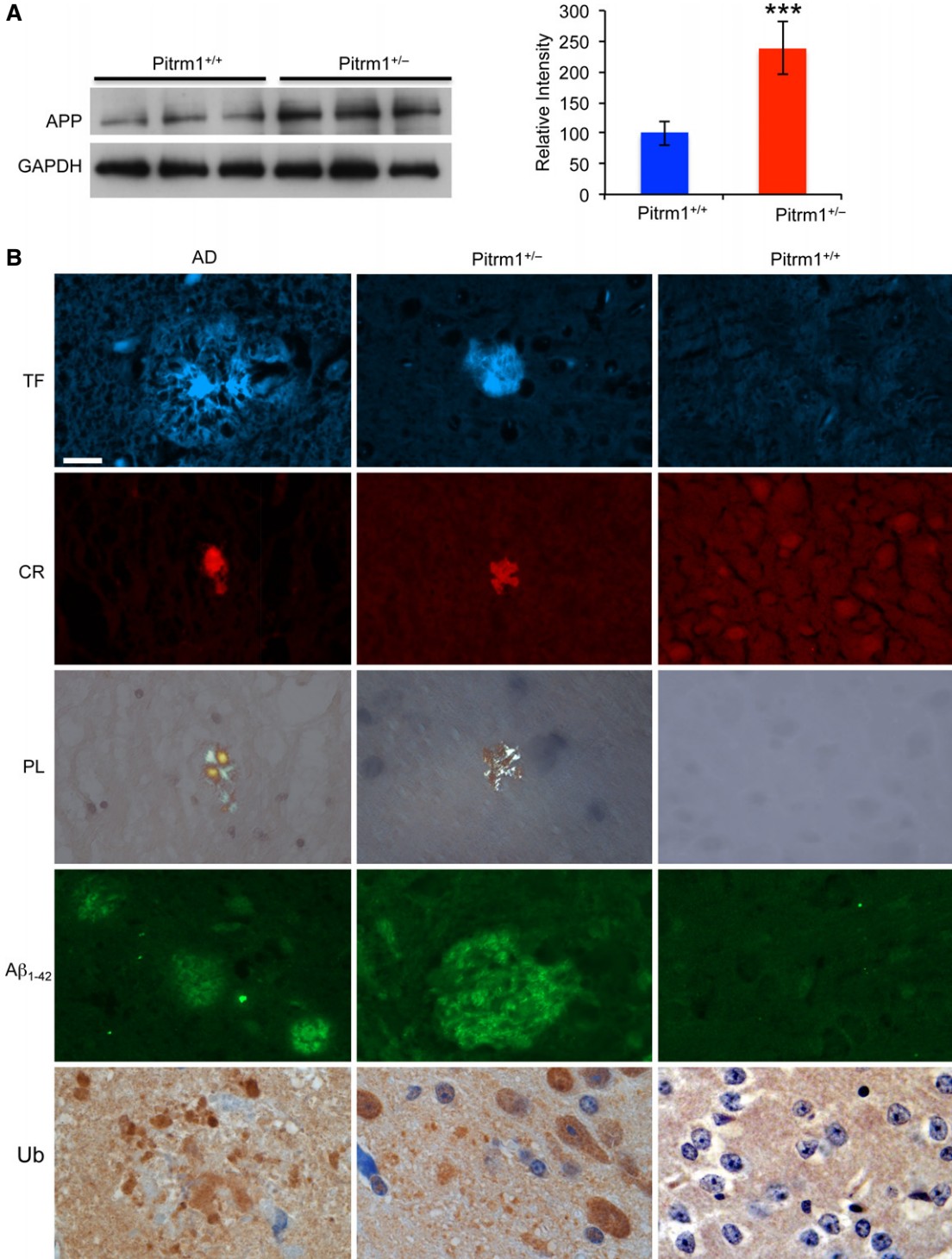

**Figure 5.  Molecular and morphological analysis of *Pitrm1*⁺/⁻ mouse brain.**

A    Western blot analysis of amyloid precursor protein (APP) in 6-mo male mice. GAPDH was used as a loading control. Densitometric analysis on a total of five independent samples is shown in the histogram on the right. ***$P$ = 0.00058. Data are presented as mean ± SD. Statistical analysis was performed using unpaired, two-tailed Student's *t*-test.

B    Morphological analysis of an AD subject and of *Pitrm1*⁺/⁻ and *Pitrm1*⁺/⁺ mouse brains. TF: thioflavin T (thalamus); CR: Congo red (brain cortex); PL: polarised light (same sections as those stained with CR); Aβ$_{1-42}$ immunostaining (pons); Ub: ubiquitin immunohistochemistry shown as brownish staining (brain cortex). Scale bar indicates 20 μm.

Source data are available online for this figure.

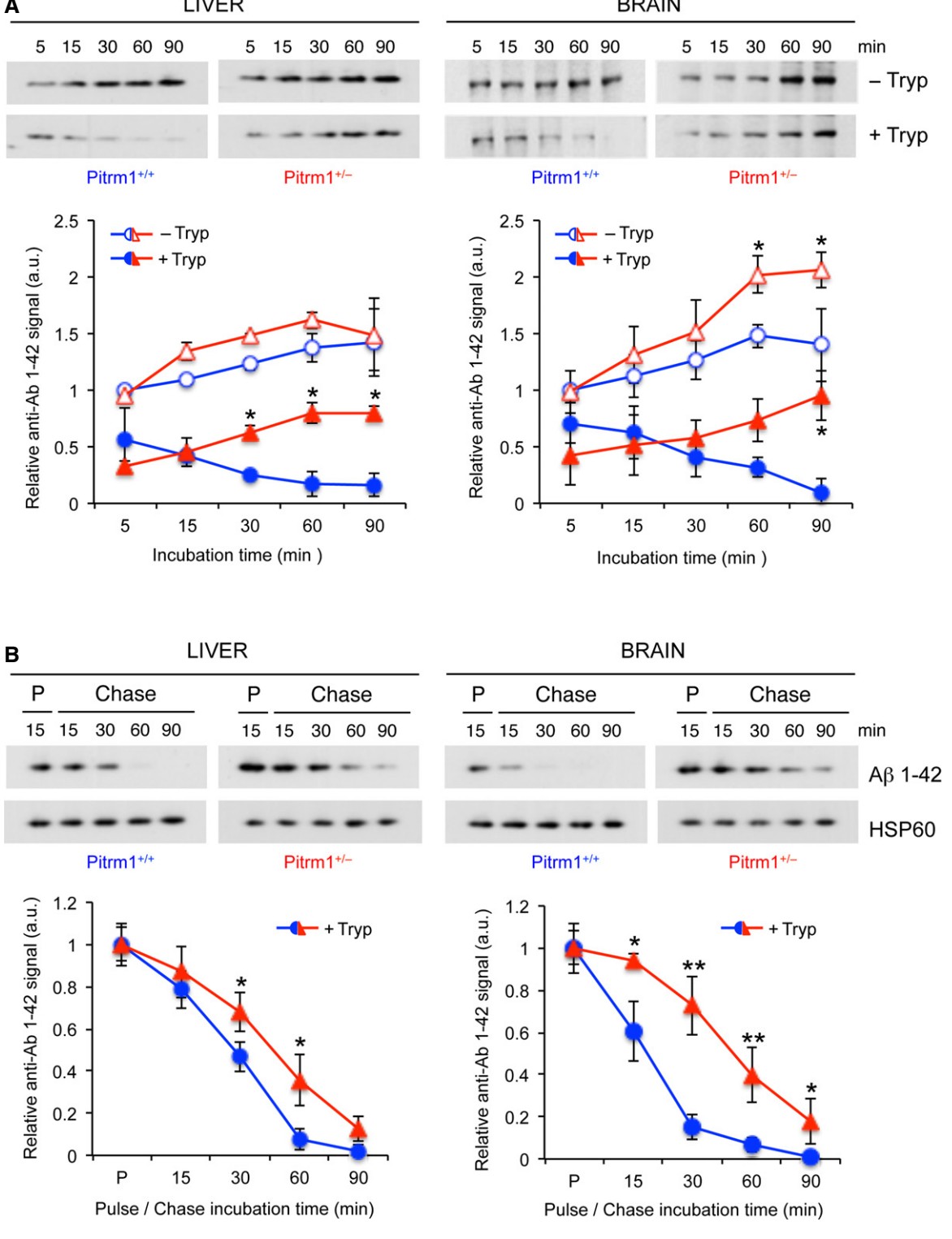

**Figure 6.  Aβ$_{1-42}$ mitochondrial import and degradation.**

A  Upper panel: import of Aβ$_{1-42}$ into mitochondria. Lower panel: relative quantification from two independent experiments. Values of Aβ$_{1-42}$ signal were normalised to HSP60 signal, and the resulting value at 5 min without trypsin was arbitrarily set as 1.

B  Upper panel: pulse and chase experiment to assess clearance of Aβ$_{1-42}$; lower panel: quantification from three independent experiments. Values of Aβ$_{1-42}$ signal were normalised to HSP60 signal, and the pulse value was arbitrarily chosen as 1.

Data information: Data are presented as mean ± SD. Statistical analysis was performed using unpaired, two-tailed Student's *t*-test. *$P < 0.05$, **$P < 0.01$. Exact *P*-values are reported in Table EV1.

(Appendix Fig S5). However, a band migrating in the region where the MTS is expected to localise was detected in $Pitrm1^{+/-}$ mitochondria in significantly higher amount than in $Pitrm1^{+/+}$ mitochondria. In support that this band was indeed the TFAM MTS stands the observation that no band corresponding to the average MTS size (approximately 3–5 kDa) was detected in both $Pitrm1^{+/+}$ and $Pitrm1^{+/-}$ mitochondria in the experiments carried out with radiolabelled MPV17. This result suggests that impaired Pitrm1 activity causes the accumulation of free MTS species *in vivo*.

### Fibroblasts from $Pitrm1^{+/-}$ mice and patients have reduced capacity to degrade Aβ peptides

Our data provide strong evidence that reduced Pitrm1 activity is associated with accumulation of Aβ plaques in the $Pitrm1^{+/-}$ brain, possibly in an age-dependent manner. This suggests that Aβ does enter mitochondria and that the activity of PITRM1 is biologically important for its removal.

To further explore this issue, we exposed to a fluorescent-labelled $Aβ_{1–40}$ peptide a number of cell lines, including immortalised $PITRM1^{R183Q}$ fibroblasts from subject II-2, normal human immortalised fibroblasts, $Pitrm1^{+/-}$ mouse embryonic fibroblasts (MEFs), control MEFs and MEFs from mitofusin 1 or 2 ko lines ($Mfn1^{-/-}$, $Mfn2^{-/-}$). Mfn1 and Mfn2 are two proteins of the outer membrane of mitochondria, both promoting mitochondrial fusion, whereas Mfn2 also mediates interactions between mitochondria and the endoplasmic reticulum (ER) (Mishra & Chan, 2014). They were included in the experiment as "positive controls" to test whether Aβ accumulation could be associated with mitochondrial fragmentation or with reduced contacts between mitochondria and the ER. After 18 h of incubation, fluorescent-labelled $Aβ_{1–40}$ was almost completely removed in normal and $Mfn1^{-/-}$ or $Mfn2^{-/-}$ controls, whereas it persisted at highly significant levels in $Pitrm1^{+/-}$ MEFs and $PITRM1^{R183Q}$ (Fig 7A), as quantitatively ascertained by fluorescent cell sorting (Fig 7B). Taken together, these results clearly demonstrate that reduced levels of PITRM1 due to either haploinsufficiency or destabilising mutations significantly impair the disposal of $Aβ_{1–40}$.

## Discussion

We have shown that Aβ accumulation participates in the neurodegeneration and neurological derangement seen in the $Pitrm1$ mouse model and, possibly, in our $PITRM1^{R183Q}$ mutant subjects as well. Aβ plaques were, however, sparse in the brain of young $Pitrm1^{+/-}$ mice, suggesting that the disease mechanism may not be limited to Aβ pathology. The main role of PITRM1 is deemed to be the elimination of cleaved mitochondrial targeting peptides after protein translocation (Koppen & Langer, 2007). Typically, these peptides, located at the N-terminus of mitochondrion-targeted proteins, are amphiphilic species, with a polar, positively charged, arginine-rich side, opposite to an apolar side (Roise & Schatz, 1988). The electrostatic features of these peptides allow them to both guide the insertion of the precursor proteins into the TIM23 translocon and drive their internalisation within the inner mitochondrial compartment, by exploiting the electrostatic component ($\Delta\Psi$) of the mitochondrial $\Delta P$ (Roise & Schatz, 1988). Due to their amphiphilic nature, however, these peptides, when released from the mature proteins

by MMP, may act as detergent-like, toxic agents, forming pores in the membranes and resulting in uncoupling and dissipation of $\Delta P$ (van 't Hof *et al*, 1991; Zardeneta & Horowitz, 1992; Nicolay *et al*, 1994). Such toxicity may explain the embryonic lethality associated with the complete ablation of PITRM1 activity in the $Pitrm1^{-/-}$ mouse genotype. Furthermore, accumulation of cleaved signal peptides may also affect maturation of mitochondrial proteins by inhibiting the mitochondrial matrix peptidase, resulting in decreased activity or instability of imported, but unprocessed, proteins (Mossmann *et al*, 2014). The same backlogging mechanism, due to impaired degradation machinery of oligopeptides, can determine reduced rate of APP processing and explain its accumulation, as detected in the brains of $Pitrm1^{+/-}$ animals.

In summary, we have identified a human neurodegenerative disease combining impaired motor coordination, cognitive and psychotic features, caused by a hypomorphic pathogenic mutation in the mitochondrial protease PITRM1 associated with protein instability. A heterozygous $Pitrm1^{+/-}$ mouse model replicated several of the neurological symptoms found in humans, and showed the presence, amongst other neuropathological features, of Aβ aggregates similar to AD amyloid plaques. Our findings offer a mechanistic link between mitochondrial dysfunction and misfolded protein aggregates in the pathogenesis of neurodegeneration. Future work is warranted to test whether PITRM1 variants are associated with amyloidotic neurodegeneration, including AD. Interestingly, the observation that the hemizygous $Pitrm1^{+/-}$ mouse displays slowly progressive, multisystem neurological impairment suggests that not only recessive variants, but also dominant or sporadic loss-of-function mutations in $PITRM1$ could be associated with adult-onset neurodegeneration, possibly characterised, as in the mouse, by accumulation of APP and Aβ deposits.

In conclusion based on medical genetic evidence (a mutant family), a yeast model, *in vitro* assays and a mouse recombinant model, we demonstrate that partial impairment of a metallopeptidase contained within the inner compartment of mitochondria not only causes neurodegeneration, but is clearly associated with accumulation of amyloid precursor protein APP and $Aβ_{1–42}$.

This wealth of independent observations leads to important conclusions, for both basic biological knowledge and translational research. First, our results conclusively resolve a long-standing debate about the presence of Aβ within mitochondria, a hypothesis that has so far remained controversial. Second, we provide genetic evidence that confirms that PITRM1 is the peptidase specifically dedicated to Aβ clearance within mitochondria and that even partial impairment of this function, caused by either instability of a mutant variant (our family) or hemizygosity (the $Pitrm1^{+/-}$ mouse), can determine neurodegeneration with accumulation of amyloidotic Aβ, directly linking the latter with abnormal mitochondrial proteostasis. This has potentially relevant implications in the etiopathogenesis of AD, a prominent cause of chronic neurological disability in the Western world. Finally, the embryonic lethality of the $Pitrm1^{-/-}$ genotype underscores the essential role of the PITRM1 protein in cellular homoeostasis and, together with the proof that PITRM1 acts on both Aβ and MTS clearance, gives mechanistic support to the idea that conditions characterised by either reduced PITRM1 activity (as in our family and mouse model) or increased Aβ production (e.g. chromosome 21 trisomy) can cause saturation of the clearance pathway centred on PITRM1, resulting in accumulation of toxic peptides,

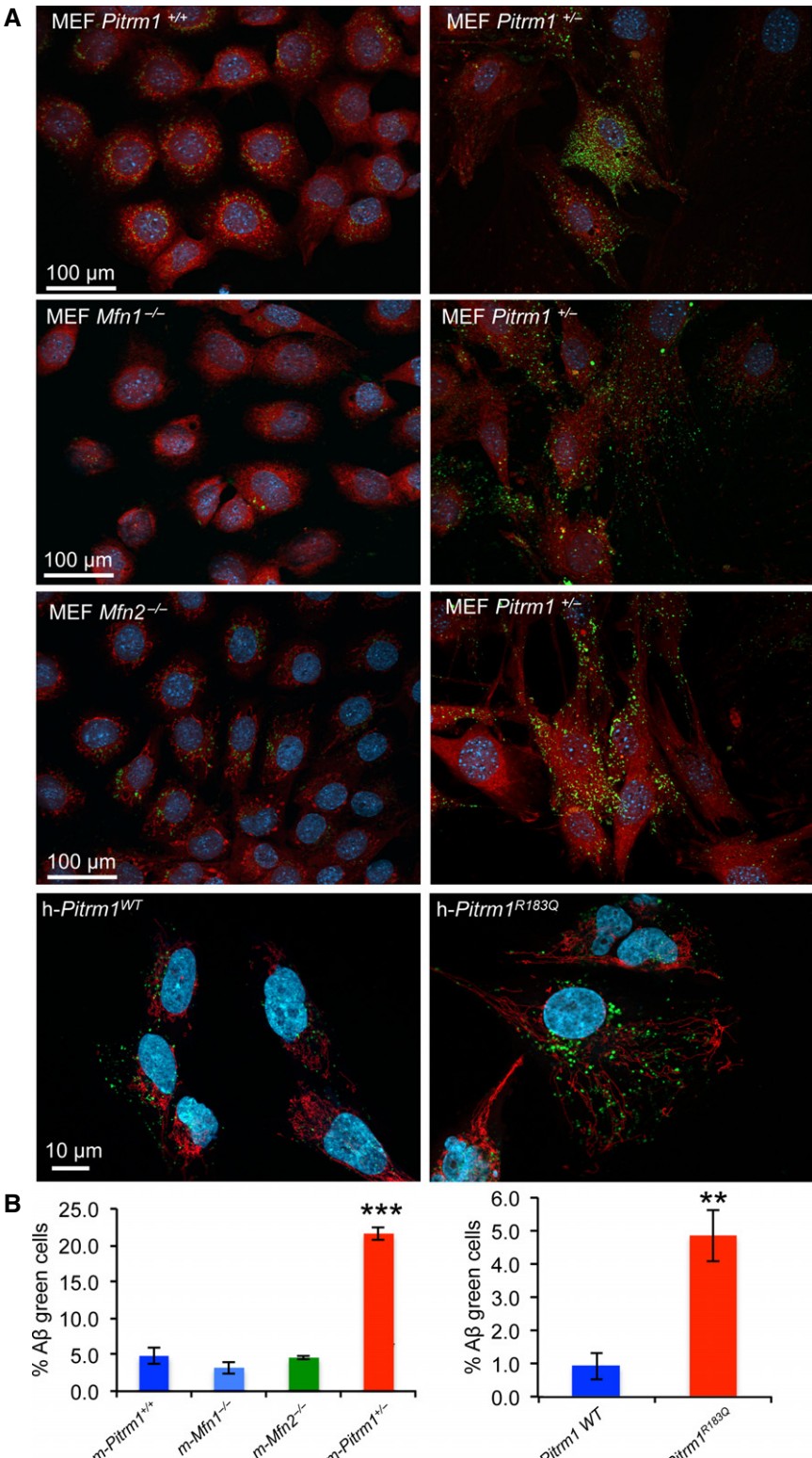

**Figure 7.    Degradation of Aβ$_{1-40}$ peptide in fibroblasts and MEFs.**

A    MEFs from *Pitrm1*$^{+/+}$, *Pitrm1*$^{+/-}$, *Mfn1*$^{-/-}$ and *Mfn2*$^{-/-}$, as well as *h-PITRM1*$^{R183Q}$ and h- *PITRM1*$^{WT}$, were exposed to Aβ$_{1-40}$ peptide for 18 h. Note that Aβ$_{1-40}$ signal was still evident in *Pitrm1*$^{+/-}$ and *h-PITRM1*$^{R183Q}$ cells.

B    Quantification of three independent experiments similar to those shown in (A). Data are presented as mean ± SD. Statistical analysis was by unpaired, two-tailed Student's *t*-test. **P < 0.01, ***P < 0.001. Exact *P*-values are reported in Table EV1.

including free MTS' and Aβ, as well as backlogging of precursors, for example APP and mitochondrial unprocessed proteins, which can eventually lead to progressive brain derangement.

# Materials and Methods

Additional methods for DNA and RNA purification, cell cultures, Western blot, biochemical assays, yeast and mouse studies are available in the Appendix Supplementary Methods.

## Genetics

Our study was approved by the Regional Committee for Medical and Health Research Ethics, Western Norway (2014/330/REK vest). Informed consent was obtained from the subjects and all the experiments conformed to the WMA Declaration of Helsinki.

Whole exome sequencing was performed at HudsonAlpha Institute for Biotechnology (Huntsville,AL) using Roche-NimbleGen Sequence Capture EZ Exome v2 kit and paired-end 100 nt sequencing on the Illumina HiSeq. The reads were mapped using v6.2 aligner, PCR duplicates removed with Picard v1.118 (http://broadinstitute.github.io/picard), and the alignment refined using Genome Analysis Toolkit (GATK) v3.2-2 and called using GATK HaplotypeCaller requiring a minimum coverage of 8 reads, and 5 reads for the variant allele. Filtering and annotation of variants were done in ANNOVAR (Haugarvoll *et al*, 2013). Coding and putative splice sites (defined as 2 bps flanking coding exons) were filtered against variants with MAF > 0.8% in an in-house database of more than 300 Norwegian exomes, and variants present at > 0.5% allele frequency in the 1000 Genomes database (phase 1 release v3 called from 20101123 alignment).

## Mitochondrial membrane potential

Detection of mitochondrial transmembrane potential ($\Delta P$) change was performed using the lipophilic, cationic dye JC-1 (ChemoMetec) and Nucleo Counter NC-3000 according to manufacturer instructions. In healthy cells with high mitochondrial $\Delta P$, JC-1 forms complexes known as J-aggregates that show an intense red fluorescence. On the contrary in unhealthy cells with low $\Delta P$, JC-1 remains in the monomeric form, showing only green fluorescence. Red vs. green fluorescence intensity ratio was quantified, and differences between cell lines were analysed statistically using two-way ANOVA.

## Indirect immunofluorescence

Fibroblasts were seeded on glass coverslips and grown for 48 h. The cells were fixed for 30 min with 3% paraformaldehyde (PFA) in 0.1 M phosphate buffer, pH 7.2 at room temperature and stained as described (Sannerud *et al*, 2008).

## *In vitro* protein modelling

### Purification of recombinant human PITRM1
Production and purification of PITRM1 (wild-type and R183Q variant) were performed as previously described (Teixeira *et al*, 2012). The PITRM1$^{R183Q}$ variant was constructed by site-directed

mutagenesis using the QuikChange II kit (Agilent Technologies) and appropriate primers and confirmed by sequencing.

### PITRM1 activity
For the analysis of Aβ degradation, PITRM1 samples (wt or R183Q, 1 μg) were incubated with 1 μg of Aβ 1–42 (Alexotech) for the indicated time in degradation buffer (50 mM HEPES-KOH pH 8.2, 10 mM MgCl$_2$) at 37°C (experiments performed in duplicate). After incubation, the reactions were resolved on NuPAGE 4–12% Bis-Tris gels and stained with Coomassie brilliant blue (Sigma).

In the fluorescence-based assays, PITRM1 samples (wt or R183Q, 0.2 μg) were mixed with either 1 μg Substrate V (sequence RPPGFSAFK, R&D Systems) or 4 μg F$_1$β 43–53 presequence fragment (sequence KGFLLNRAVQYK, custom synthesis), and the increase in fluorescence (excitation 327 nm; emission 395 nm) was recorded on a plate reader (SpectraMax Gemini). Experiments were performed in duplicate, and the results are shown as substrate degradation rates.

## Yeast studies

Strains and oligos used in this work are reported in Appendix Tables S1 and S2, respectively. Strain W303-1B *cym1Δ* was obtained by one-step gene disruption with a KanMX4 cassette amplified from the corresponding BY4742 deleted strain, using primers CYM1DCFw and CYM1DCRv. All experiments, except transformation, were performed in synthetic complete (SC) media (0.19% YNB without amino acids and NH$_4$SO$_4$ powder (ForMedium, Norfolk, UK), 0.5% NH$_4$SO$_4$) supplemented with 1 g/l dropout mix without amino acids or bases necessary except those necessary to keep plasmids. Media were supplemented with various carbon sources as indicated below (Carlo Erba Reagents, Milan, Italy) in liquid phase or after solidification with 20 g/l agar (ForMedium). Growth was performed with constant shaking at 37°C. Transformation with suitable recombinant plasmids was used to express CYM1$^{wt}$ and cym1$^{R163Q}$ protein variants, each carrying an HA epitope on the C-terminus for immunovisualisation. Additional details are reported in Appendix Supplementary Methods.

## Mouse studies

All animal experiments were carried out in accordance with the UK Animals (Scientific Procedures) Act 1986 and EU Directive 2010/63/EU for animal experiments. The C57BL/6n-A$^{tm1Brd}$ *Pitrm1*$^{+/−}$ mice used in this study were kindly provided by the Sanger Institute (http://www.informatics.jax.org/ allele/MGI:5085349). Animals were housed two or three per cage in a temperature-controlled (21°C) room with a 12-h light–dark cycle and 60% relative humidity. The experimental design included two groups of male mice of 3–6 months of age (9 *Pitrm1*$^{+/+}$ vs. 7 *Pitrm1*$^{+/−}$).

### Morphological analysis of mouse brain
Histological and immunohistochemical analyses were performed on formalin-fixed and paraffin-embedded brain tissues. Five-μm-thick serial sections were stained with haematoxylin–eosin and viewed by light microscopy. For histochemical studies, tissues were frozen in liquid-nitrogen precooled isopentane and serial 8-μm-thick sections were stained for COX, SDH and NADH as described (Sciacco &

Bonilla, 1996). Congo red and Thioflavin T stainings were performed as described (Puchtler & Sweat, 1962; Burns *et al*, 1967). Congo red-positive areas were viewed under polarised light, while Thioflavin T staining was evaluated by fluorescence microscopy.

### Mitochondrial import and degradation of Aβ

For import experiments, $A\beta_{1-42}$, human TFAM and MPV17 radio-labelled proteins were obtained via coupled transcription and translation (TNT) in a reticulocyte system in the presence of [$^{35}$S]-metionine. Liver and brain mitochondria were isolated by differential centrifugation (Reyes *et al*, 2011) and incubated with 0.36 mg/ml $A\beta_{1-42}$, for 5 to 90 min in import buffer (Petruzzella *et al*, 1998). Then, half of the reaction was treated with trypsin for 15 min, and mitochondrial pellets were resolved on 10–20% PAGE gels. $A\beta_{1-42}$ and HSP60, used as loading control, were analysed by Western blot.

For degradation experiments with $A\beta_{1-42}$, liver and brain mitochondria were incubated with 0.36 mg/ml $A\beta_{1-42}$ for 15 min (pulse) followed by trypsin digestion and incubation for 15 to 90 min (chase).

### Exposure of human and mouse cells to fluorescent Aβ

MEFs ($Pitrm1^{+/+}$ and $Pitrm1^{+/-}$) and human immortalised fibroblast ($PITRM1^{WT}$ and $PITRM1^{R183Q}$) were grown for 24 h on glass slides and then exposed for 18 h to fluorescent $A\beta_{1-40}$ peptide ($A\beta_{1-40}$ HiLite fluor 488-labelled, AnaSpec), freshly dissolved in PBS and added to culture medium at a final concentration of 1 μM for 18 h at 37°C. Subsequently, the medium was changed and MitroTracker Red (Invitrogen) was added for 30 min at 37°C. Cells were washed with PBS and fixed with 2% PAF for 20 min. Cells were washed with PBS and mounted using Prolong Gold antifade reagent with DAPI. The samples were visualised by an inverted laser scanning microscope (Axio Observer.Z1).

### Statistics

Unless differently indicated in the figure legends, statistical analysis was performed using unpaired, two-way Student's *t*-test. Data are presented as mean ± SD. Exact *P*-values for all experiments are reported in Table EV1.

**Expanded View** for this article is available online.

### Acknowledgements

We thank the Wellcome Trust Sanger Institute Mouse Genetics Project (Sanger MGP) and its funders for providing the mutant mouse line. Funding and associated primary phenotypic information may be found at http://www.sanger.ac.uk/science/collaboration/mouse-resource-portal. Our work is supported by the following grants: Cariplo 2011-0526, ERC FP7-322424 (to M.Z.); the Swedish Research Council (to EG); Helse Vest 911810 (PMK); Forening for muskelsyke (to LAB); and GR-2010-2306-756 from the Italian Ministry of Health (to CV). We are grateful to: Charalampos Tzoulis for help with figures, Hrvoje Miletic for the immunohistochemistry figure Roberta Ruotolo and Simone Ottonello for pYES2-mtβ₄₂ vector; Alan Robinson for the bioinformatics analysis; Ian Fearnley and Mike Harbour for the proteomics; Sabrina Ravaglia for advice in some statistical analyses; and Jorunn Bringsli for technical assistance; the personnel at ARES and Phenomics Animal Care Facilities for the support in managing our colonies. This paper

**The paper explained**

**Problem**

Two siblings of a consanguineous family with a slowly progressive neurodegenerative disorder of unknown origin were investigated to establish the cause.

**Results**

A homozygous, disease-segregating missense mutation was found in the *PITRM1* gene in both siblings. The pathogenic role of the mutation, causing PITRM1 instability, was validated by *in vitro* assays, characterisation of mutant fibroblasts from patients, in *PITRM1* knocked-down human fibroblasts and in a mutant yeast model. A hemizygous PITRM1 knockout mouse displayed reduced amounts of PITRM1 and slowly progressive neurodegeneration, characterised by accumulation of amyloid beta (Aβ) in the brain.

**Impact**

We have identified a clinically peculiar human neurodegenerative disorder caused by a pathogenic, homozygous mutation in *PITRM1*, a gene encoding an oligopeptidase of the mitochondrial inner compartment. The neuropathology of a $Pitrm1^{-/+}$ mouse provides genetic evidence that Aβ is present within mitochondria, and demonstrates a link between impaired PITRM1 activity and Aβ amyloidotic neurodegeneration in mammals.

is dedicated to the memory of Gottfried Schatz, PhD and Stefano di Donato, MD.

### Author contributions

JT performed laboratory investigations of patient samples. WT was involved in ascertaining and following the patient and organising clinical investigations. PS, SJ and HB were responsible for SNP chip analysis and WES and data interpretation. PT and EG were responsible for generating recombinant PITRM1 and testing activity. DB characterised the $Pitrm1^{+/-}$ mouse, and performed part of the experiments on mouse and human cells. AR performed the import and pulse and chase experiments on mouse cells and isolated mitochondria. CD, EB, PG and IF designed and performed the experiments on the yeast models, and EF-V performed and supervised most of the experiments on human cell lines and tissues and contributed to those on isolated mitochondria. RC, CP and GD performed and interpreted the histochemical and histological data. CV supervised the work on the mouse model. MZ and LAB designed the study and wrote the manuscript.

### Conflict of interest

The authors declare that they have no conflict of interest.

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
