## [Review Process File · EMBO Molecular Medicine]

Defective PITRM1 mitochondrial peptidase is associated with A amyloidotic neurodegeneration

Dario Brunetti, Janniche Torsvik, Cristina Dallabona, Pedro Teixeira, Pawel Sztromwasser, Erika Fernandez-Vizarra, Raffaele Cerutti, Aurelio Reyes, Carmela Preziuso, Giulia d'Amati, Enrico Baruffini, Paola Goffrini, Carlo Viscomi, Ileana Ferrero, Helge Boman, Wenche Telstad, Stefan Johansson, Elzbieta Glaser, Per Knappskog, Massimo Zeviani, and Laurence A. Bindoff

Corresponding authors: Massimo Zeviani, Medical Research Council Mitochondrial Biology Unit and Laurence A. Bindoff, University of Bergen

Review timeline:	Submission date:	26 September 2015
	Editorial Decision:	21 October 2015
	Revision received:	29 October 2015
	Editorial Decision:	11 November 2015
	Revision received:	16 November 2015
	Editorial Decision:	17 November 2015
	Revision received:	19 November 2015
	Accepted:	23 November 2015

Editor: Roberto Buccione

Transaction Report:

1st Editorial Decision	21 October 2015
-----------------

Thank you for the submission of your manuscript to EMBO Molecular Medicine. We have now heard back from the three Reviewers whom we asked to evaluate your manuscript.

Although all Reviewers are globally positive on your manuscript, a number of issues are raised that require further action. I will not dwell into much detail, but I would like to highlight the main points.

Reviewer 1 is critical of the mouse studies and is not convinced of their actual relevance and is especially critical of the quality and presentation of the morphology/neuropathology findings. In general s/he laments many missing details that are crucial to ensure reproducibility, a topic that is quite close to our hearts at EMBO Press. This Reviewer also suggests that some important aspects should be properly discussed (e.g. origin of the A β deposits, uptake of APP and A β by mitochondria, etc.) and points to a number of unsupported statements.

Reviewer 2 is less reserved but would like you to improve the clinical presentation by clearly discussing the differences with bona fide Alzheimer's disease. Connected also to Reviewer 1's concerns, it would be quite useful and informative to present a comparative analysis vs. mouse models of Alzheimer's disease.

Reviewer 3 has a number of very interesting questions, including on the genetics of the disease, on the muscle phenotype and on etiopathology. I suggest that you address these carefully.

In conclusion, while publication of the paper cannot be considered at this stage, we would be pleased to consider a revised version of your manuscript with the understanding that the Reviewers' concerns must be addressed and that acceptance of the manuscript will entail a second round of review.

I look forward to seeing a revised form of your manuscript as soon as possible.

***** Reviewer's comments *****

Referee #1 (Comments on Novelty/Model System):

The authors report a family with a homozygous missense mutation in the PITRM1 gene that encodes the mitochondrial matrix protein pitrilysin metallopeptidase 1 (PITRM1). PITRM1 is presequence peptidase that degrades presequences of proteins targeted to the matrix. The homozygous missense mutation R183Q was found in two individuals with a slowly progressing neurodegenerative phenotype. The gene and the missense mutation were identified by SNP-based mapping and whole exome sequencing, and confirmed by Sanger sequencing. This is an important, novel finding which connects a mitochondrial mutation to neurodegeneration.

Information on the index case and her brother are provided in the text - a table would be more appropriate - and relate to age, mental retardation beginning in childhood, development of spinocerebellar ataxia (SCA), obsessional behavior with psychotic episodes and hallucination. Brain imaging of the two cases showed marked cortical and cerebellar atrophy. No quantitative figures on the atrophy are provided. According to what is mentioned in the text, the index case showed low A β levels of 363ng/L, and it is mentioned that this is similar to what is seen in idiopathic Alzheimer's disease (AD). However, this is not correct since only A β 42 is lower in AD, not A β 40 and not total A β . Which A β is referred to and how A β levels were determined is not specified. This needs to be done. Total CSF tau, phospho tau and 14-3-3 protein levels were normal suggesting that the PITRM1 mutation does not cause AD neuropathology.

Interestingly, no abnormality was found for respiratory chain complexes in muscle homogenates from the index case - probably, because these activities are not associated with the mitochondrial matrix.

Investigations of skin fibroblasts and skeletal muscle biopsy taken from the index case by qPCR showed unimpaired PITRM1 RNA expression but Western-blot analyses showed strong reduction of PITRM1 protein in both tissues. By expressing PITRM1 in *E. coli*, the authors could show that the mutation does not impair catalytic activity suggesting that the R183Q missense mutation impaired protein stability. These studies address the structural implications of the mutations and are well documented in Fig. 1.

Further experiments with immortalized fibroblasts from the index patient and controls showed a growth defect on galactose medium in fibroblasts derived from the patient and a 40% reduction of PITRM1 protein levels in control fibroblast treated with PITRM1-specific shRNA. Which shRNA has been used is not mentioned but should have been included to allow repetition of the experiment by others.

Studies of the *S. cerevisiae* PITRM1 orthologue CYM1 were done in CYM1 null mutants after transfection with the corresponding *cym1*R163Q allele and allowed to confirm that the mutation leads to a reduced amount of *cym1*R163Q protein compared to CYM wt *S. cerevisiae*. Furthermore, expression of A β 1-42 fused to the mitochondrial signal peptide of Sod2 showed complete degradation in CYM1 wt but no degradation in delta *cym1* and incomplete degradation in *cym1* R163 Q. These findings suggest that PITRM1 regulates intramitochondrial A β turnover.

Experimental details are given for the generation of the CAM1 null mutant but not for its transfection. The latter is required to allow repetition of the experiments.

Mouse studies were performed with C57 BL/6n-Atm 1Brd *Pitrm1* +/- mice. Since these mice are not models for the human cases with PITRM1R183Q missense mutations, the experiments with these mice are a compromise. The ideal mouse model would be a knock in of *Pitrm1* with the equivalent of the PITRM1R183Q missense mutation. The finding that *Pitrm1* levels are reduced by 50-60% in different tissues of the *Pitrm1* +/- mice compared to *Pitrm1* 1+/+ mice is trivial and does not add to the understanding of the pathogenic effects in humans. The same holds true for the sensorimotor and coordination tests, especially since no details on these tests are given in Methods. The increases in APP in brain homogenates of this mouse model as well as the APP immunohistochemistry in postmortem brains of the *Pitrm1* +/- mouse are not addressed in the discussion, which should have

been done. The neuropathological studies presented in Fig 5B do also not add to the understanding of the pathogenesis as long as the brain regions shown are not specified and no information is provided on the prevalence of the lesions shown in Fig. 5B.

Studies on degradation of A β peptides in immortalized fibroblast from the index patient showed that fluorescent-labeled A β 1-40 persisted at higher levels in these cells but not in normal controls. The same was reported for MEF's derived from the Pitrm1 +/- mice and again does not add new information to what has been found in the human fibroblasts. The legend to Fig. 7A is in regard to the MEF panels incomplete and should explain why three panels are shown. Some if not all of the mouse data should be moved to the Supplement.

Major claims:

The authors claims are as following. The first and most important claim is the identification of two siblings carrying a homozygous missense mutation in the PIRTM1 gene, which encodes the mitochondrial matrix enzym pitrilysin metallopeptidase 1 (PITRM1). Second, the identification of the missense mutation in PIRTM1 exchanges the arginine residue 183 that is conserved in humans and yeast to a glutamine residue. The third claim relates to the hypothesis that the homozygous missense mutation is associated with an autosomal recessive mental retardation syndrome, spinal cerebellar ataxia, cognitive decline and psychosis and that this phenotype was partly reproduced in C57 BL/6n-Atm 1Brd Pitrm1 +/- mice. The forth claim relates to the finding that the PITRM1R183Q missense mutation lowers PITRM1 protein levels but not PIRTM1 enzymatic activity and thus leads to the attenuation of degradation of peptides containing the mitochondrial matrix targeting signal sequence and of A β , and leads to A β accumulation in mitochondria. All these claims are novel. The majority of the claims are convincing and supported by experiments with the exception of those experiments where no or insufficient experimental details are given.

As mentioned already, the discussion does not address the findings on APP and A β increases in the C57 BL/6n-Atm 1Brd Pitrm1 +/- mice. Where do the parenchymal A β deposits come from in these mice? Why are APP brain levels increased? What are the possible mechanisms and what is known about the uptake of APP and A β by mitochondria? What is known in this regard of the role of mitochondrial membrane potential (FCCP experiments) and of HSP 60? All these questions have not been addressed experimentally but should be discussed at least.

The statement made within the first sentence of the discussion is not supported by the experiments and at best a working hypothesis. This needs to be clarified.

The conclusions at the end of the discussion stating that instability of a familial variant of PITRM1 can determine amyloidotic A β -based neurodegeneration, directly linking the latter with abnormal mitochondrial proteostasis is again a working hypothesis and not yet a fact. The same holds true for the title of the paper that needs to be changed. The title should express what has been found, namely, that a familial variant of PITRM1 mitochondrial peptidase is associated with accumulation of mitochondrial A β and neurodegeneration.

Referee #1 (Remarks):

same as above under 4.

Referee #2 (Comments on Novelty/Model System):

The neurologists are expert in the field and the mitochondrial studies are carried out by world leading expert on mitochondria pathophysiology. The study cover investigations from patients to transgenic animals.

Referee #2 (Remarks):

The paper reports a new genetic syndrome characterized by mental retardation, psychosis and spinocerebellar ataxia and caused by mutation responsible for misfolding and early degradation of the mitochondrial enzyme pitrilysin .

The authors have discovered that the absence of the enzyme significantly alters the Abeta metabolism and the patients have a low level of Abeta in CSF such as occurs in Alzheimer patients.

In the mouse model the lack of pitrilysin causes the accumulation of Abeta as amyloid plaques and causes a neurodegenerative syndrome

The discovery of a link between the syndrome and the abnormal metabolism is remarkable and will have great impact on further studies of the effect of Abeta on mitochondrial metabolism and toxicity. The study is carried out at the state of the art with all the possible experimental approaches for characterizing the role of the mitochondrial enzyme on Abeta metabolism.

The clinical pathological study is of course limited by ethical issues and most likely the lack of tissues and limited access to serial MRI data.

The authors should better clarify that the clinical features are not classical for an Alzheimer disease. Based on the description of the clinical features I have the impression that dementia is not the main component of the syndrome. The data regarding the low concentration of Abeta (used as a biomarker of Alzheimer), if not properly discussed, can confuse the reader on the pathologic basis of the syndrome. I'm wondering if the neurologist would better discuss the putative role of classical neurodegeneration caused by plaques of Abeta on the clinical aspects of the syndrome.

Minor comment

New pictures of the congo red staining are required because the putative and diagnostic green birefringence after polarization cannot be appreciated in the actual picture

Referee #3 (Comments on Novelty/Model System):

The paper by Brunetti et al. is an excellent study, reporting a new mitochondrial disease caused by mutations in the PITRM1 gene. They detected a homozygous point mutation in the mitochondrial matrix enzyme PITRM1, which resulted in childhood onset mild mental retardation, followed by spinocerebellar ataxia, cognitive decline and psychosis. The authors performed a very nice and complex set of experiments in human cells, yeast and mice in support of their hypothesis, that PITRM1 provides a link between mitochondria and amyloidotic neurodegeneration. This is a very exciting novel finding!

Referee #3 (Remarks):

The paper by Brunetti et al. is an excellent study, reporting a new mitochondrial disease caused by mutations in the PITRM1 gene in a Norwegian family. The disease pathomechanism is very interesting; the mutation in the mitochondrial matrix enzyme PITRM1 results in abnormal accumulation of cleaved mitochondrial targeting sequences and also of Amyloid beta. The clinical presentation is childhood onset mild mental retardation, which is followed by spinocerebellar ataxia, cognitive decline and psychosis. The authors performed a complex set of relevant experiments in human cells, yeast and mice in support of their hypothesis, that PITRM1 suggests a link between mitochondria and amyloidotic neurodegeneration. This is a very exciting novel finding!

I have only a few questions:

1. Both cleaved MTSs and Amyloid beta are increased in this disease. Do the authors suggest a causal relationship between these two, or are they just rather coexisting effects of the defect of PITRM1 mutations?
2. I find it interesting that the disease already causes mild mental retardation in childhood and later on other symptoms of neurodegeneration (ataxia, psychosis, cognitive decline) develop. What could be the cause of the early onset symptoms? Has there been any imaging performed earlier in the disease course or only after age 60 years?
3. What is the suggested mechanism of the ragged blue fibres? If citrate synthase is low, it would indicate a lower number of mitochondria.
4. The authors performed mtDNA deletion/depletion analysis in immortalised fibroblasts of a patient, which did not show decreased mtDNA copy number and also did not detect a significant number of mtDNA deletions. Was it also tested in skeletal muscle DNA of the patient? These abnormalities are often only present in affected post-mitotic tissues (such as skeletal muscle).
5. Based on the finding of a phenotype in the heterozygous mice the authors point out an important point, that although in the reported family the mutation in PITRM1 is homozygous, this disease may be dominant due to other mutations in this gene. Even heterozygous nonsense mutations with

haploinsufficiency may cause a phenotype. Did the authors try to look for additional families in Alzheimer disease or ataxia cohorts?

1st Revision - authors' response

29 October 2015

We are grateful to the editor and referees for their comments regarding our manuscript. We felt that these were on the whole very positive. All 3 reviewers described the findings as “important” and/or “novel” and understood the clear connection between a mutation involving a purely matrix mitochondrial protease and a form of neurodegeneration that includes amyloid deposition. This is hugely important to us since this is the first time that this has been shown. We hope moreover, that the first reviewer also shares our and the other two referees' view on the great importance of this finding and does not let (a few) technical omissions cloud the issue.

For the benefit of discussion we would reiterate:

We state clearly that the knock-out mouse is embryonically lethal. Further, the major detectable effect of the homozygous human missense mutation is a reduction in the amount of the PITRM1 protein, we show that the catalytic activity is maintained, making the hemizygous *-/+* mouse model a very good approximation of the human condition. The main conceptual advancement provided by the mouse model is to show a clear link between A β metabolism, mitochondrial proteostasis and neurodegeneration. Validation of the pathogenic role of the human mutation is otherwise demonstrated by a massive amount of experimentation carried out in human cells from patients, with RNAi models in human cells, as well as by in vitro experiments using recombinant protein, and in a highly informative yeast model. Lastly, as stated in our discussion, the finding that hemizygous mice do develop a neurological phenotype raises the possibility of finding dominantly transmitted or sporadic heterozygous PITRM1 mutations causing human disease. For all these reasons we feel that our study, including the mouse KO model, the yeast and in vitro studies and the detailed evaluation of patient cells combine altogether to make a coherent and important advance in our understanding of human neurodegeneration.

Specific comments

Reviewer 1.

1. *This is an important, novel finding which connects a mitochondrial mutation to neurodegeneration*
We entirely agree.
2. *Informations on the index case and her brother are provided in the text - a table would be more appropriate*
The provision of clinical data is a matter of style; we would of course change it were it made an editorial requirement.
3. *Brain imaging of the two cases showed marked cortical and cerebellar atrophy. No quantitative figures on the atrophy are provided*
Quantification requires multiple scans and this would need to be clinically indicated. If we were making a specific point concerning volume loss, it would be more appropriate to include numerical measurements. In this case, where the scans are important to demonstrate the range of clinical involvement, we feel that visual evaluation should be sufficient to convince one that there is atrophy.
4. *Only Ab1-42 is lower in AD, not Ab1-40 and not total A β . Which Ab is referred to and how Ab levels were determined is not specified. This needs to be done*
We agree that we should have been more specific here. These are clinical tests performed in accredited laboratories. We have now clarified that the measurement refers to Ab₁₋₄₂.
5. *Which shRNA has been used is not mentioned but should have been included to allow repetition*
Again, we apologise for the omission of the technical data. We have now added the full details: Five different MISSION® shRNA lentiviral constructs targeting human PITRM1 mRNA: TRCN0000052238-42, were purchased from Sigma-Aldrich. All five were tested in control, immortalized fibroblasts, being TRCN0000052238, TRCN0000052241 and TRCN0000052240 the most efficient, in this order, in knocking-down PITRM1 expression.

6. *Experimental details are given for the generation of the CYM1 null mutant but not for its transfection. The latter is required to allow repetition of the experiments*

The technique used to transform yeast has been added in Experimental procedures section at the end of the “Yeast studies” paragraph and in Extended Methods in the Supplemental Material Online with additional details

7. *The ideal mouse model would be a knock in of Pitrm1 with the equivalent of the PITRM1R183Q missense mutation*

We are afraid we disagree with this statement. A knock-in mouse could be interesting, but this in no way reduces the interest or importance of the heterozygote KO mouse. As we state above, the KO mouse faithfully reproduces the major finding in humans, namely the PITRM1 protein instability. Whether this would occur (or not) with a knock-in model, is a moot point, however, to state that – “the finding that *Pitrm1* levels are reduced by 50-60% in different tissues of the *Pitrm1* +/- mice compared to *Pitrm1* +/+ mice is trivial and does not add to the understanding of the pathogenic effects in humans” suggests to us that the reviewer has not fully given a pondered thought to the biological nature and potential medical relevance of our findings.

8. *The same holds true for the sensory-motor and coordination tests, especially since no details on these tests are given in Methods*

We invite the Referee to take a good look at the Supplementary Material, where, for space reasons, we thought convenient to report in detail large parts of the most commonly used methods. Has the referee perhaps overlooked this. However, we have now added a sentence to make clear that additional methods are provided separately

9. *The increases in APP in brain homogenates of this mouse model as well as the APP immunohistochemistry in postmortem brains of the Pitrm1 +/- mouse are not addressed in the discussion, which should have been done*

A mechanism to explain the accumulation of APP is contained in the very last sentence of the Discussion. However, we have now added a further sentence in the discussion to propose that “The same backlogging mechanism, due to impaired degradation machinery of oligopeptides can determine a reduced rate of APP processing and explain its accumulation, detected in the brains of *Pitrm1*^{+/-} animals”

10. *The neuropathological studies presented in Fig 5B do also not add to the understanding of the pathogenesis as long as the brain regions shown are not specified and no information is provided on the prevalence of the lesions shown in Fig. 5B*

The figure has been completed with the requested labeling.

11. *Studies on degradation of Ab peptides in immortalized fibroblast from the index patient showed that fluorescent-labeled Ab1-40 persisted at higher levels in these cells but not in normal controls. The same was reported for MEF's derived from the Pitrm1 +/- mice and again does not add new information to what has been found in the human fibroblasts. The legend to Fig. 7A is in regard to the MEF panels is incomplete and should explain why three panels are shown. Some if not all of the mouse data should be moved to the Supplement.*

We cannot agree with this point of view. The studies on human fibroblasts in addition to MEF's from the KO mouse are fundamental to confirming that failure of amyloid removal is not a mouse specific occurrence, and at the same time they confirm that the mouse is a good model for what happens in the patients. We will of course remedy any, if any, lack of information in the legend, but suggest that this figure remains in the main text, as it provides relevant information pertinent to the main topic of the paper.

12. *As mentioned already, the discussion does not address the findings on APP and Ab increases in the C57 BL/6n-Atm 1Brd Pitrm1 +/- mice. Where do the parenchymal Ab deposits come from in these mice? Why are APP brain levels increased? What are the possible mechanisms and what is known about the uptake of APP and Ab by mitochondria? What is known in this regard of the role of mitochondrial membrane potential (FCCP experiments) and of HSP60? All these questions have not been addressed experimentally but should be discussed at least.*

The statement made within the first sentence of the discussion is not supported by the experiments and at best a working hypothesis. This needs to be clarified.

It is difficult to understand what this reviewer wishes us to do in the discussion. It is convention to constrain the discussion to what is relevant to the results presented, with, of course, some consideration of the broader issues raised. In our experience, most editors would treat any tendency to overelaborate harshly. We are presenting absolutely novel data

demonstrating that amyloid beta accumulates in the presence of a mitochondrial matrix enzyme defect. Currently we cannot say where the deposits come from or why APP levels are increased, although we suggest a "backlogging" effect that has also been shown to occur in yeast by previous papers. These are surely the subject for future studies when we can examine in detail the mechanisms involved.

The statement made in the first paragraph appears to us to be quite reasonable: we have shown that in the Pitrm^{+/-} mouse, A β accumulates, that there is neurodegeneration and neurological signs. We feel it quite realistic to say as well that we presume the same is happening in the patients. We must emphasize, however, that the pathogenetic mechanism leading to neurodegeneration must be elucidated in its full mechanistic details. We think it is unlikely that the scattered Ab deposits found in the Pitrm1^{-/+} brains can per se determine the neurological phenotype observed in Pitrm1^{-/+} mice. Our working hypothesis is that reduced activity of the Pitrm1 oligopeptidase causes overall derangement of mitochondrial proteostasis, accumulation of toxic oligopeptides, and reduced availability of mitochondrial mature proteins. The accumulation of Ab in this context is one of the effects, not the only one, of Pitrm1 failure, but stands as genetic proof of the role of mitochondrial protein quality control on the metabolism of Ab.

Reviewer 2

1. *The discovery of a link between the syndrome and the abnormal metabolism is remarkable and will have great impact on further studies of the effect of Abeta on mitochondrial metabolism and toxicity.*

We agree and thank this reviewer for the comment.

2. *The authors should better clarify that the clinical features are not classical for an Alzheimer disease. Based on the description of the clinical features I have the impression that dementia is not the main component of the syndrome.*

We are not sure we understand the point raised by this reviewer. We have not made any suggestion that our patients had AD. The reviewer is correct, however, in assuming that dementia was not the major component. The disorder started early and cognitive development was not normal. The patients were already >50 so much of their early course was difficult to elicit.

3. *New pictures of the congo red staining are required because the putative and diagnostic green birefringence after polarization cannot be appreciated in the actual picture*

We have changed the relevant panels in Figure 4 and added an Extended View Figure to further support the evidence that Ab accumulates in the brains of -/+ mice.

Reviewer 3

The authors performed a very nice and complex set of experiments in human cells, yeast and mice in support of their hypothesis, that PITRM1 provides a link between mitochondria and amyloidotic neurodegeneration. This is a very exciting novel finding!

We are grateful for these comments

Both cleaved MTSs and Amyloid beta are increased in this disease. Do the authors suggest a causal relationship between these two, or are they just rather coexisting effects of the defect of PITRM1 mutations?

This is an interesting question. In our in vitro studies, we show that the R183Q causes no loss of enzyme activity. The accumulation of both MTS's and amyloid beta is presumed to be the result of PITRM1 protein instability causing haplo-insufficiency.

The results in mice support this since the KO has lower PITRM1 levels than controls.

What is the suggested mechanism of the ragged blue fibres? If citrate synthase is low, it would indicate a lower number of mitochondria.

The fibres are COX negative, but do not show any significant mitochondrial accumulation (raggedness). It is difficult to comment on the significance in someone >60; they could indicate a moderate OXPHOS defect linked to the underlying genetic defect or just be age-related.

The authors performed mtDNA deletion/depletion analysis in immortalised fibroblasts of a patient, which did not show decreased mtDNA copy number and also did not detect a significant number of mtDNA deletions. Was it also tested in skeletal muscle DNA of the patient? These abnormalities are often only present in affected post-mitotic tissues (such as skeletal muscle).

Unfortunately we haven't carried out this analysis and the muscle biopsy is no longer available. We decided to mention the essentially negative results on mtDNA depletion/deletion in fibroblasts as "not shown" to improve the readability of Figure 2

Based on the finding of a phenotype in the heterozygous mice the authors point out an important point, that although in the reported family the mutation in PITRM1 is homozygous, this disease may be dominant due to other mutations in this gene. Even heterozygous nonsense mutations with haploinsufficiency may cause a phenotype. Did the authors tried to look for additional families in Alzheimer disease or ataxia cohorts
We entirely agree and have instituted a search. We felt, however, that the data reported here as of sufficient interest and importance to make a stand alone publication.

2nd Editorial Decision

11 November 2015

Thank you for the submission of your revised manuscript to EMBO Molecular Medicine. We have now received the enclosed reports from the referees that were asked to re-assess it. As you will see the reviewers are now globally supportive. I also asked Reviewer #3 to assess your replies to Reviewer #1: S/he feels that you have also answered the comments of the other reviewer correctly, and completely agrees with your views. I am thus pleased to inform you that we will be able to accept your manuscript pending the following final amendments:

1) As per our Author Guidelines, the description of all reported data that includes statistical testing must state the name of the statistical test used to generate error bars and P values, the number (n) of independent experiments underlying each data point (not replicate measures of one sample), and the actual P value for each test (not merely 'significant' or 'P < 0.05').

2) We are now encouraging the publication of source data, particularly for electrophoretic gels and blots, with the aim of making primary data more accessible and transparent to the reader. Would you be willing to provide a PDF file per figure that contains the original, uncropped and unprocessed scans of all or at least the key gels used in the manuscript? The PDF files should be labeled with the appropriate figure/panel number, and should have molecular weight markers; further annotation may be useful but is not essential. The PDF files will be published online with the article as supplementary "Source Data" files. If you have any questions regarding this just contact me.

3) Every published paper now includes a 'Synopsis' to further enhance discoverability. Synopses are displayed on the journal webpage and are freely accessible to all readers. They include a short standfirst as well as 2-5 one sentence bullet points that summarise the paper. Please provide the synopsis including the short list of bullet points that summarise the key NEW findings. The bullet points should be designed to be complementary to the abstract - i.e. not repeat the same text. We encourage inclusion of key acronyms and quantitative information. Please use the passive voice. Please attach this information in a separate file or send them by email, we will incorporate it accordingly. You are also welcome to suggest a striking image or visual abstract to illustrate your article. If you do please provide a jpeg file 550 px-wide x 400-px high.

4) Please upload the appendix directly as a pdf, not word file.

***** Reviewer's comments *****

Referee #2

Comments on Novelty/Model System

The revised version does not require any change

Remarks

The revised version clarify the few issues raised by this reviewer

Referee #3

Comments on Novelty/Model System

I find the answers of the corresponding author informative and satisfactory and I have no more comments.

Remarks

The revised version clarify the few issues raised by this reviewer